# Mortality Risk and Urinary Proteome Changes in Acute COVID-19 Survivors in the Multinational CRIT-COV-U Study

**DOI:** 10.3390/biomedicines12092090

**Published:** 2024-09-13

**Authors:** Justyna Siwy, Felix Keller, Mirosław Banasik, Björn Peters, Emmanuel Dudoignon, Alexandre Mebazaa, Dilara Gülmez, Goce Spasovski, Mercedes Salgueira Lazo, Marek W. Rajzer, Łukasz Fuławka, Magdalena Dzitkowska-Zabielska, Harald Mischak, Manfred Hecking, Joachim Beige, Ralph Wendt

**Affiliations:** 1Mosaiques Diagnostics GmbH, 30659 Hannover, Germany; siwy@mosaiques-diagnostics.com (J.S.);; 2Department of Internal Medicine IV (Nephrology and Hypertension), Medical University of Innsbruck, 6020 Innsbruck, Austria; 3Department of Nephrology and Transplantation Medicine, Wrocław Medical University, 50-556 Wroclaw, Poland; 4Department of Molecular and Clinical Medicine, Institute of Medicine, The Sahlgrenska Academy at University of Gothenburg, 413 45 Gothenburg, Sweden; 5Department of Nephrology, Skaraborg Hospital, 541 85 Skövde, Sweden; 6Department of Anaesthesiology and Critical Care, Saint Louis-Hôpital Lariboisière, AP-HP, 75010 Paris, France; 7Department of Epidemiology, Medical University of Vienna, 1090 Vienna, Austria; 8Department of Nephrology, University Sts. Cyril and Methodius, 1000 Skopje, North Macedonia; 9Virgen Macarena Hospital, University of Seville, 41009 Seville, Spain; 10First Department of Cardiology, Interventional Electrocardiology and Arterial Hypertension, Jagiellonian University Medical College, 30-688 Kraków, Poland; 11Department of Clinical and Experimental Pathology, Wroclaw Medical University, 50-556 Wrocław, Poland; 12Molecular Pathology Centre Cellgen, 50-353 Wrocław, Poland; 13Faculty of Physical Education, Gdańsk University of Physical Education and Sport, 80-336 Gdańsk, Poland; 14Centre of Translational Medicine, Medical University of Gdańsk, 80-210 Gdańsk, Poland; 15Kuratorium for Dialysis and Transplantation (KfH) Leipzig, 04129 Leipzig, Germany; 16Department of Internal Medicine II, Martin-Luther-University Halle-Wittenberg, 06120 Halle, Germany; 17Department of Nephrology, St. Georg Hospital Leipzig, 04129 Leipzig, Germany

**Keywords:** long COVID, PASC, biomarker, mortality, peptides, urine

## Abstract

Background/Objectives: Survival prospects following SARS-CoV-2 infection may extend beyond the acute phase, influenced by various factors including age, health conditions, and infection severity; however, this topic has not been studied in detail. Therefore, within this study, the mortality risk post-acute COVID-19 in the CRIT-COV-U cohort was investigated. Methods: Survival data from 651 patients that survived an acute phase of COVID-19 were retrieved and the association between urinary peptides and future death was assessed. Data spanning until December 2023 were collected from six countries, comparing mortality trends with age- and sex-matched COVID-19-negative controls. A death prediction classifier was developed and validated using pre-existing urinary peptidomic datasets. Results: Notably, 13.98% of post-COVID-19 patients succumbed during the follow-up, with mortality rates significantly higher than COVID-19-negative controls, particularly evident in younger individuals (<65 years). These data for the first time demonstrate that SARS-CoV-2 infection highly significantly increases the risk of mortality not only during the acute phase of the disease but also beyond for a period of about one year. In our study, we were further able to identify 201 urinary peptides linked to mortality. These peptides are fragments of albumin, alpha-2-HS-glycoprotein, apolipoprotein A-I, beta-2-microglobulin, CD99 antigen, various collagens, fibrinogen alpha, polymeric immunoglobulin receptor, sodium/potassium-transporting ATPase, and uromodulin and were integrated these into a predictive classifier (DP201). Higher DP201 scores, alongside age and BMI, significantly predicted death. Conclusions: The peptide-based classifier demonstrated significant predictive value for mortality in post-acute COVID-19 patients, highlighting the utility of urinary peptides in prognosticating post-acute COVID-19 mortality, offering insights for targeted interventions. By utilizing these defined biomarkers in the clinic, risk stratification, monitoring, and personalized interventions can be significantly improved. Our data also suggest that mortality should be considered as one possible symptom or a consequence of post-acute sequelae of SARS-CoV-2 infection, a fact that is currently overlooked.

## 1. Introduction

The life expectancy of SARS-CoV-2 survivors remains a subject of critical inquiry, affected by various factors such as age, pre-existing health conditions, and the severity of the initial infection [1]. While much attention has been directed toward understanding the acute phase of COVID-19, there is a growing recognition of the long-term health implications for those who have recovered. It has become increasingly evident that the impact of SARS-CoV-2 extends beyond the immediate symptoms experienced during the acute phase of the disease. The COVID-19 pandemic has markedly elevated global mortality [2], with a significant proportion of COVID-19 survivors facing lingering health challenges and increased morbidity in the months and years following recovery from acute infection.

Among those at risk for severe outcomes, older individuals with pre-existing health conditions such as cardiovascular disease, diabetes, and respiratory disorders are particularly vulnerable [3,4]. Studies have shown that male sex and ethnicity—specifically Black or South Asian backgrounds—as well as the severity of the initial infection correlate with higher rates of COVID-19-related death [5]. These factors contribute to a complex interplay of risks that affect the long-term prognosis of COVID-19 survivors. Some patients experience persisting or new developing disease burdens after the acute SARS-CoV-2 infection that cannot be attributed to any alternative diagnosis. This phenomenon is also documented after other acute respiratory infections but to a lesser extent [6]. Common symptoms include fatigue, shortness of breath, cognitive dysfunction, and reduced ability to perform daily activities of life, and new symptoms are also being observed [7]. The overlapping conditions and the variable onset of symptoms were recently reviewed in detail [8]. In October 2021, the World Health Organisation established the first clinical case definition for long COVID (LC), which was recently redefined [9,10,11,12]. Given the estimate that 10% of persons with SARS-CoV-2 acquire LC, there are at least 65 million individuals around the world affected [8]. Vaccination has been shown to reduce the incidence of LC [13,14]. Because of the absence of any biomarkers for LC, the diagnosis is almost entirely reliant on reported symptoms and questionnaires [15]. Investigations in large cohorts show a large variety of new or persisting symptoms in up to 45% of COVID-19 survivors [16]; however, correcting for individual pre-existing symptoms and comparing with symptom dynamics in COVID-19-negative populations during the pandemic leads to a much lower prevalence of LC [17]. The average duration of LC symptom clusters is estimated to be 9.0 months for hospitalized individuals and 4.0 months for non-hospitalized individuals, with 15.1% who continue to experience symptoms at 12 months [12]. Women are clearly more affected [12].

The pathophysiology of LC remains insufficiently understood. Persistent SARS-CoV-2 in the body has been considered a potential factor [18,19]. Patients with persisting infection showed more than 50% higher odds of self-reporting LC [20]. The COVID-19 duration was also associated with cognitive deficits after recovery [21]. Additionally, reduced serotonin levels may contribute to the neurocognitive symptoms associated with viral persistence in LC [22]. Persisting immunological dysfunction in LC patients might also play an important role [23]. Further analyses revealed significant heterogeneity and large biological diversity in LC with clusters and subsets with distinct signatures, reaching from persistent inflammation to non-inflammatory LC [23,24]. Multidimensional immune phenotyping identified discriminating biological features associated with LC [25]. Despite the growing body of research on acute COVID-19 and LC, the long-term mortality among COVID-19 survivors is poorly understood, with specific underlying mechanisms remaining elusive. It is hypothesized that the prolonged effects and ongoing physiological stress and organ damage caused by persistent COVID-19 symptoms could contribute to an increased risk of mortality, especially among those with pre-existing conditions or those who experienced severe acute infection [26]. This study seeks to address this knowledge gap by examining mortality rates among patients who have survived the acute phase of the illness and by identifying specific urinary peptides that may be associated with future death. Urinary peptides are short sequences of amino acids that can reflect pathological processes in the body, and their identification could potentially serve as biomarkers for predicting long-term health outcomes in COVID-19 survivors.

Staessen et al. in the study “*Prospective Validation of a Proteomic Urine Test for Early and Accurate Prognosis of Critical Course Complications in Patients with SARS-CoV-2 Infection*” (CRIT-COV-U) investigated urinary peptides in 1012 adults with PCR-confirmed COVID-19 [27]. The research focused on the acute phase of the disease with a median follow-up of 10 days. The authors demonstrated that it is possible to predict adverse COVID-19 outcomes within the acute phase of the illness using specific urinary peptides, and this prediction can be made within 4 days of a positive PCR test [27]. Within this study, mortality rates among patients who have survived the acute phase of COVID-19 are investigated to identify potential predictors of long-term post-acute COVID-19 mortality (PACM), including demographic factors, clinical characteristics, and laboratory parameters. Additionally, the role of urinary peptides as biomarkers of future death is examined, leveraging recent advances in proteomic technology to explore their predictive value. Detection of urinary peptides associated with future mortality could have significant implications for risk stratification and personalized care among COVID-19 survivors. 

Incorporating these biomarkers into clinical practice could allow healthcare providers to better identify individuals at high risk of adverse outcomes and tailor interventions accordingly. This would ultimately improve long-term outcomes and quality of life for survivors of COVID-19, contributing to more effective management of the pandemic. By integrating advanced biomarker detection with traditional clinical assessments, the ability to predict, monitor, and manage the long-term health impacts of COVID-19 can be enhanced, ensuring that survivors receive the comprehensive care they need to mitigate future health risks.

## 2. Materials and Methods

### 2.1. Study Population

Post-acute survival data of 651 unvaccinated patients were gathered from the “*Prospective Validation of a Proteomic Urine Test for Early and Accurate Prognosis of Critical Course Complications in Patients with SARS-CoV-2 Infection*” (CRIT-COV-U) study, spanning until December 2023 across six countries and nine centers (Appendix A) [27]. These patients were enrolled during the initial and subsequent waves of the pandemic in 2020–2021, predominantly infected with the wild-type virus, and had survived the acute phase of COVID-19. Urine peptide data of first urine samples collected within 3 days of a positive PCR were used. The cohort was stratified into discovery (n = 324) and validation (n = 327) sets through random partitioning. This project complied with the Helsinki Declaration. The Ethics Committee of the German–Saxonian Board of Physicians (Dresden, Germany; number EK-BR-70/23-1) and the Institutional Review Boards of the recruiting sites provided ethical approval. To assess the impact of age on mortality within this cohort, comparisons were made against age- and sex-matched data from individuals not infected with SARS-CoV-2 (n = 5192), sourced from the Human Urinary Database [28].

### 2.2. Urinary Peptidomics

Data were extracted from the Human Urinary Proteome Database, which contains datasets acquired using capillary electrophoresis coupled with mass spectrometry (for details on the CE-MS analysis, please see [29]) as described previously [28]. Data were evaluated using MosaFinder software (version 1.4) and normalized based on the abundance of 29 collagen peptides [30]. Of the 5071 sequenced peptides identified to date, only those present in at least 50% of the entire discovery cohort of 324 individuals (923 peptides) were retained for further analyses.

### 2.3. Statistical Analysis

As descriptive statistics for the samples, the median and interquartile range (IQR) were used for continuous variables and absolute (N) and relative frequencies (%) for categorical variables. Hypotheses of no differences in scale or distribution of patient characteristics between the death and non-death groups were tested with Wilcoxon–Mann–Whitney test for continuous variables and with χ2 homogeneity tests for categorical variables. Adjustment for multiple testing was implemented according to Benjamini [31,32]. Visualized are kernel density estimates of the distribution of the scores split by mortality groups. Mortality per person–time stratified by age and DP201 groups, is estimated as the ratio of the number of the deceased to the sum of all patients’ observation times within each group scaled to 100 person-years. The corresponding mortality probabilities and their 95% confidence intervals (CI) for each group represent estimates from a logistic regression including all 651 patients.

### 2.4. Classifier Development

A classifier combining multiple features (peptides) into a single variable was developed using support vector machine modeling as described in [33]. All peptides demonstrating a significant difference (adjusted for the false-discovery rate set at 0.05) between cases and controls were included in the classifier. Classification was performed by determining the Euclidian distance (classification score) of the vector to a separating hyperplane. The optimal parameters for C (cost of misclassification) and gamma (flexibility of the separating hyperplane) were determined via leave-one-out cross-validation error estimation, as described in more detail in [34]. 

## 3. Results

### 3.1. Assessment of Mortality in Acute COVID-19 Survivors

Of the 893 patients from the CRIT-COV-U study surviving acute COVID-19, follow-up data from 651 patients could be obtained (Table 1). At the time of inclusion in the CRIT-COV-U study (and urine sampling), the median age of the 651 patients was 63 years (IQR: 48–76)), with a male predominance of 53.5%. The median body mass index (BMI) recorded was 27.0 (IQR: 24.4–30.3) kg/m^2^, and the estimated glomerular filtration rate (eGFR) was 90.0 (IQR: 70–111) mL/min/1.73 m^2^. The majority of patients, 56.4%, had no recorded comorbidities. The entry WHO scores were 1–3 in 311 (48%) participants, 4–5 in 317 (49%) participants, and 6 in 23 (4%) participants. Throughout the follow-up period, spanning a median of 2.92 years (IQR: 2.67–3.09), pertinent data were collected to assess mortality outcomes in survivors of the acute phase of COVID-19.

Among the 651 patients who survived the acute phase of COVID-19 and could be followed up on, 91 individuals (13.98%) succumbed during the follow-up duration, with 55 (8.45%) of these fatalities occurring within the first year post-infection. In stark contrast, among the age- and sex-matched controls totaling 5192 individuals, a markedly lower proportion of 92 (1.77%) deaths were recorded within the same time frame. Notably, mortality displayed an age-dependent pattern across both cohorts, with significantly elevated rates observed among those who had survived COVID-19 compared to their COVID-19-negative counterparts (Figure 1A–E). Specifically, within the first year post-infection, mortality rates surged up to 4.7 times higher in patients younger than 65 years compared to the COVID-19-negative controls.

### 3.2. Identification of Biomarkers Associated with Post-Acute COVID-19 Mortality

For the identification of biomarkers potentially associated with mortality after surviving the initial acute phase of COVID-19, the previously acquired urinary peptidomics datasets from baseline samples of COVID-19-diagnosed patients within the CRIT-COV-U study were utilized. These datasets were stratified into discovery (n = 324) and validation cohort (n = 327) sets through random partitioning. Urinary peptides potentially associated with PACM were defined by applying the Mann–Whitney test to compare 44 deceased and 280 surviving patients within the discovery set. Subsequently, adjustments for multiple testing were implemented to ensure statistical robustness.

The analysis of urinary peptidome datasets within the discovery set enabled the identification of 201 peptides (listed in Appendix A) as significantly associated with PACM when comparing deceased and surviving patients. These peptides encompassed upregulated fragments of albumin, alpha-2-HS-glycoprotein, apolipoprotein A-I, and beta-2-microglobulin, alongside downregulated fragments of CD99 antigen, various collagens, fibrinogen alpha, polymeric immunoglobulin receptor, sodium/potassium-transporting ATPase, and uromodulin. Among these peptides, 14 overlapped with the previously established Cov50 classifier designed for prognosticating unfavorable COVID-19 outcomes during the acute phase [35]. 

### 3.3. Establishment and Validating a Classifier Predicting Post-Acute COVID-19 Mortality

The 201 peptides significantly associated with PACM were combined to form a support-vector machine-based classifier (DP201). This classifier enabled separating the discovery set with 80% sensitivity and 83% specificity upon complete leave-one-out cross-validation (area under the curve (AUC) = 0.86, Figure 2A). Subsequently, this classifier was applied to the independent validation cohort, consisting of 47 deceased and 280 surviving patients, which resulted in significant separation of the groups with an AUC of 0.78, as shown in Figure 2B. 

The resultant outcomes in relation to follow-up time are depicted in Figure 3A,B, illustrating a clear correlation between higher classification scores and heightened mortality risk. Further Cox regression analysis revealed that age, BMI, and DP201 were significantly associated with PACM, while sex, number of comorbidities, eGFR, and COVID-19 WHO score did not exhibit statistical significance. Integration of these three parameters into Cox’s model yielded a hazard ratio of 6.28 (95%CI: 3.54–11.44) compared to age and DP201 alone (Figure 3C,D).

## 4. Discussion

Our findings first and foremost demonstrate that COVID-19 is associated with a highly significantly increased risk of mortality even after the acute phase of the disease. To date, this issue was apparently not well-covered by the studies, which either investigated the immediate outcome of the acute infection or the long-term effect in the context of the post-acute sequelae of SARS-CoV-2 infection (PASC). However, PASC was typically only assessed in patients still alive. Our data indicate that a significant number of patients may have died as a result of the consequences associated with the previous SARS-CoV-2 infection. The mortality in post-acute COVID-19 patients was significantly higher than in age- and sex-matched controls and the deaths could be potentially labeled as PASC-related deaths. 

A retrospective analysis of 13,638 patients with COVID-19 hospitalization documented a significantly increased risk for future mortality; increased 12-month mortality was observed in patients with severe COVID-19 compared to COVID-19-negative patients, which was concluded to be an under-investigated sequela of COVID-19 [36]. 

Another large retrospective analysis of long-term outcomes of 22,571 adult patients hospitalized due to COVID-19 in Austria in the year 2020 found an increased mortality compared to 217,295 propensity score matched controls [37]. Similar to our results, the difference between patients and controls remained significant in the younger age groups (41–64 years and 65–74 years, *p* < 0.001) but not in the oldest age group (*p* = 0.078) [37].

An investigation was conducted on the long-term risks in over 800,000 COVID-19 patients compared the risk of post-discharge death with 56,409 Influenza patients as a historical control; patients who were discharged alive from a COVID-19-related hospitalization admission had nearly twice the risk of post-discharge death compared to historical controls admitted to hospital with influenza [38]. 

Data from a large study with 47,780 English patients discharged alive after COVID-19 hospitalization showed an increased risk of readmission and mortality during a follow-up of 140 days. The post-discharge mortality risk was eight times greater than in matched controls and with the largest differences in the age group < 70 years [26]. 

A report from the US investigated the mortality after recovery from the initial episode of COVID-19 and reported a significantly higher 24-month-adjusted all-cause mortality risk for patients with severe COVID-19 compared to COVID-19-negative comparators (HR 2.01). The risk of excess death was highest during days 0 to 90 after infection (aHR 6.36) and still elevated during days 91 to 180 (aHR 1.18). Beyond 180 days after infection there was no excess mortality during the next 1.5 years [39]. A recent large US cohort of 135,161 people with SARS-CoV-2 infection and 5,206,835 controls were followed for 3 years to estimate risks of death and PASC. Among non-hospitalized individuals, the increased risk of death was no longer present after the first year of infection, while among hospitalized individuals, risk of death was high in the first year (incidence rate ratio: 3.17) and declined but remained significantly elevated even in the third year after infection (IRR 1.29) [40]. 

SARS-CoV-2 infection obviously poses a persistent threat to individuals even beyond the acute phase. Importantly, among those who successfully navigate the acute phase of COVID-19, the risk of mortality escalates significantly during the subsequent follow-up period. Particularly noteworthy is the observation that within the first year following infection, mortality rates surge dramatically among individuals who have survived the acute phase of the illness when compared to a COVID-19-negative control cohort. What is striking is that this increase in mortality risk is most pronounced among younger individuals, highlighting a concerning trend that defies conventional assumptions regarding age-related vulnerability to severe outcomes. The most abundant significantly changed peptides in patients experiencing death during follow-up are derived from β2-microglobulin (B2M). Higher B2M serum concentrations are associated with higher mortality in the general population, non-dialyzed chronic kidney disease patients, and patients receiving hemodialysis (HD) [41]. 

The data also show a consistently higher level of uromodulin peptides in patients without event. This is in very good agreement with a recent study presented by Vasquez-Rios and colleagues, where increased levels of uromodulin were found associated with a lower risk of cardiovascular death [42]

Thymosin beta4 (TB4) is an abundant actin-sequestering protein that has been described in the context of multiple (patho)physiological processes, among others, including wound healing, angiogenesis, and migration, to name just a few. It has also been described as increased in kidney disease, with the highest levels detected in patients with end-stage kidney disease [43]. Drum and colleagues found TB4 to be significantly increased in women with heart failure with preserved ejection fraction and associated with mortality. The strict association with female sex may be the result of TB4 being an X-linked gene product, which consequently is also found higher in women [44].

The Sodium/potassium-transporting ATPase subunit gamma (FYXD2) is found to be highly expressed in the kidney distal tubulus. In previous studies, a reduced abundance of peptides derived from FYXD2 has been observed as associated with the progression of CKD, specifically of IgA nephropathy [45].

Reduced levels of a peptide derived from S100A9 consequently reduced degradation of this protein and likely results in increased levels of calprotectin, which was described as associated with an increased risk of mortality [46].

Reduced abundance of peptides from the polymeric immunoglobulin (PIGR) receptor was previously found associated with acute COVID-19 mortality [35]. Similarly, increased complement activation, which result in an increase in complement-derived urine peptides, was also described as associated with increased COVID-19 mortality [47]. 

The most pronounced effect is on collagen fragments, with both an increase and decrease in specific collagen-derived peptides being observed. A change in collagen peptides has been described for multiple diseases and was also found to be associated with mortality, including mortality in the context of COVID-19 [27,48]. As observed here, both the up- and downregulation of collagen fragments were observed. This was interpreted as disruption of collagen degradation, leading to increased fibrosis. Two urinary peptide-based classifiers, CKD273 [49] and FPP_BH29 [50]—both based on multiple specific collagen peptides—were presented as highly significantly associated with fibrosis. To investigate if the changes observed in this study are associated with increased fibrosis, we applied these two classifiers onto the data and compared the scoring in the survivors vs. patients experiencing death in follow-up. As shown in Figure 4, we observe a highly significant increase in both scores, indicating increased fibrosis in the case group.

Also, a consistent reduction in CD99 was observed in severe cases of COVID-19, where a significant reduction in CD99 also was found on the surface of peripheral blood lymphocytes [51]. Based on the data, the authors hypothesized that reduction in CD99 may have a negative impact on the endothelial barrier integrity, a well-known phenomenon in severe COVID-19.

The increase in urinary albumin, fetuin, and apolipoprotein A1 may all be consequences of a similar underlying mechanism: endothelial dysfunction resulting in a loss of functionality of the glomerular filtration barrier. In fact, an increase in albuminuria is well-known and associated with an increased risk of mortality.

The increase in alpha 1 antitrypsin, a major plasma inflammatory protein, was found associated with increased mortality in the Nagahama study based on 9682 subjects [52]. This is in line with the observed increase associated with PACM in our study, which may be further exacerbated by the proteinuria, as mentioned above.

Furthermore, the identification of specific urinary peptides capable of predicting heightened mortality risk at the outset of SARS-CoV-2 infection underscores the intricate interplay between molecular biomarkers and clinical outcomes. These peptides serve as early indicators of the likelihood of mortality, providing valuable insights into the underlying pathophysiological mechanisms driving adverse outcomes in COVID-19 patients. By leveraging these predictive biomarkers, healthcare professionals can proactively identify individuals at elevated risk of mortality and implement targeted interventions aimed at mitigating this risk, thereby potentially altering the trajectory of the disease course, like already shown for chronic kidney and heart diseases [53].

A shortcoming of this study may be that the patients investigated were not immunized against SARS-CoV-2 as at the time of the initial study, the vaccine was generally not available. While mortality due to acute COVID-19 has been reduced dramatically as a result of immunization, it is not certain that immunity protects from PACM equally well, and this needs to be investigated in a subsequent study.

## 5. Conclusions

Our findings underscore the multifaceted nature of SARS-CoV-2 infection, extending far beyond the acute phase and exerting a lasting impact on mortality outcomes. The acute phase of COVID-19 appears to initiate a complex disease trajectory in many survivors. Apparently, recovery from the acute phase does not necessarily equate to a return to pre-infection health. Instead, many survivors experience ongoing health challenges, often referred to “long COVID” or PASC, which includes persistent fatigue, cardiovascular complications, cognitive impairments, and other chronic conditions. Our study adds mortality risk in COVID-19 survivors to the list of symptoms. This risk is driven by the interplay between demographic factors (such as age), pre-existing health conditions (like cardiovascular disease, diabetes mellitus, and respiratory disorders), and molecular changes, detectable in the urinary peptidome. Understanding these interactions may help in identifying individuals who are at higher risk of adverse outcomes, enabling more precise and effective healthcare interventions.

The identified predictive biomarkers represent a significant advancement in our ability to foresee and manage long-term health risks in COVID-19 survivors. These biomarkers offer a window into the biological processes that continue to affect patients long after the initial infection has resolved. The approach applied here not only enhances our understanding of the disease but also holds the promise of improving monitoring and treatment of COVID-19 survivors. By integrating predictive biomarkers into clinical practice, a more effective risk stratification and personalized interventions can be achieved. Personalized interventions, informed by a patient’s unique biomarker profile [53], can improve management strategies and optimize treatment plans, ultimately leading to better health outcomes.

This research lays the groundwork for improved clinical management and patient outcomes by providing a robust framework for predicting and mitigating long-term risks associated with SARS-CoV-2 infection, consequently offering a path toward more proactive and patient-centered care. By focusing on the long-term health of COVID-19 survivors, their quality of life can be enhanced and mortality rates could be reduced.

## Figures and Tables

**Figure 1 biomedicines-12-02090-f001:**
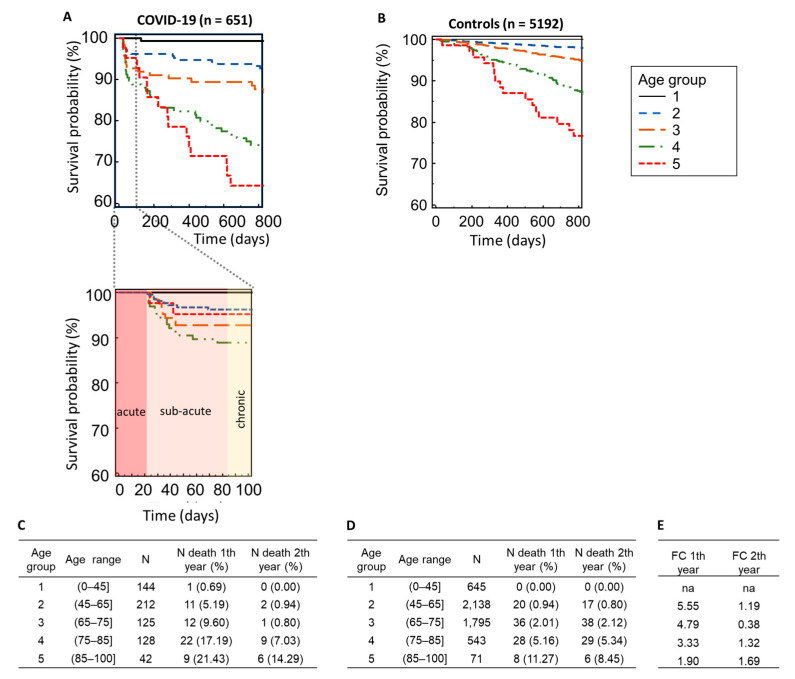
Age-dependent mortality in the post-acute COVID-19 cohort and healthy controls. Kaplan–Meier survival curves are shown for the cohort who survived acute-phase COVID-19 (**A**), including zoomed-in view of the acute (without detected deaths) and sub-acute phases, which highlight that only those who survived the acute phase were followed up, and age- and sex-matched COVID-19-negative healthy controls (**B**). The number of deaths per age group in the post-acute COVID-19 cohort (**C**) and controls (**D**) as well as the calculated fold change (FC) between the COVID-19 and controls (**E**) is given.

**Figure 2 biomedicines-12-02090-f002:**
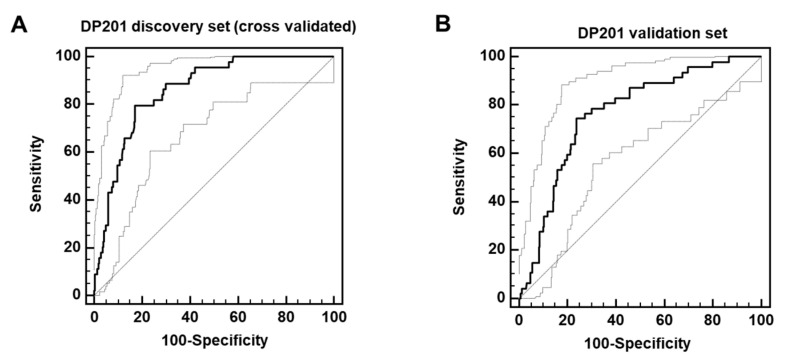
ROC curves (displayed as solid lines) for the classification of the deceased and surviving patients in the complete leave-one-out cross-validated discovery cohort (**A**) and independent validation cohort (**B**). Dotted lines display 95% confidence bounds.

**Figure 3 biomedicines-12-02090-f003:**
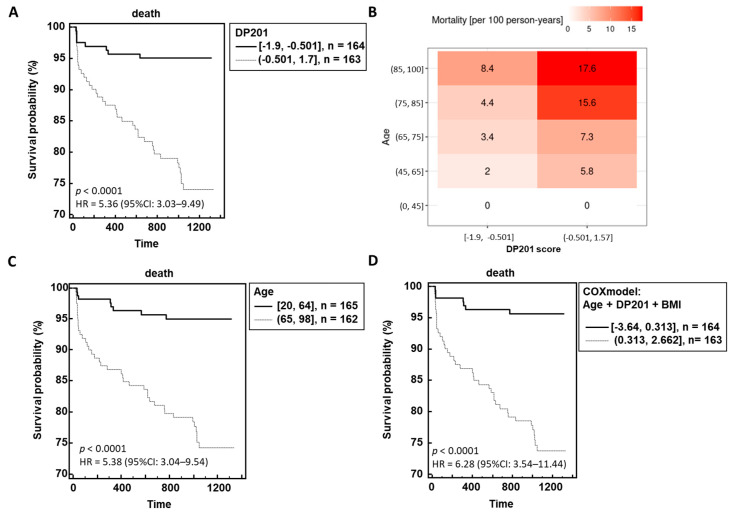
Performance of the urinary peptide-based death prediction classifier in the independent validation data. The risk of death is significant dependent on the DP201 score (**A**) although the age dependency can still be observed (**B**). The hazard ratio for survival probability DP201 classifier (**A**) and age (**C**) could be increased using a model including DP201, age, and BMI (**D**).

**Figure 4 biomedicines-12-02090-f004:**
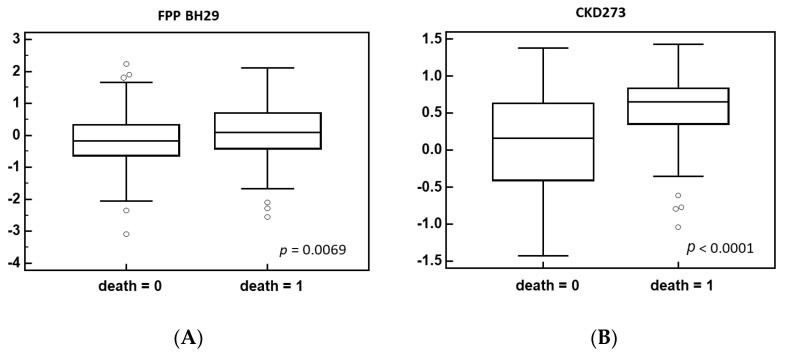
Distribution of the classification scores between the deceased and surviving patients for the fibrosis-related classifier FPP BH29 (**A**) and CKD273 classifier (**B**).

**Table 1 biomedicines-12-02090-t001:** Baseline cohort characteristics. Median and IQR are shown for continuous variables and absolute (N) and relative frequencies (%) for categorical variables.

	No Death (n = 560)	Death (n = 91)	*p*-Value
Age	60 (45–73)	78 (70–83)	<0.0001
BMI [kg/m^2^]	27.1 (24.5–30.3)	26.5 (23.7–29.9)	0.1929
Number of comorbidities	0.0 (0.0–1.0)	1 (0.3–2.0)	<0.0001
eGFR [mL/min/1.73 m^2^]	92.47 (76.00–112.17)	69 (52.00–90.00)	<0.0001
Heart rate [beats per min]	80.0 (72.0–80.0)	80.0 (70.0–86.5)	0.1827
Diastolic blood pressure [mm Hg]	78.0 (70.0–82.0)	73.0 (64.3–80.0)	0.0494
Systolic blood pressure [mm Hg]	128 (115.0–140.0)	128 (110.0–140.0.)	0.5809
sex, men (%)	289 (51.6)	58 (63.7)	0.0416
WHO score admission	3 (2–4)	4 (3–4)	<0.0001

## Data Availability

The data that support the findings of this study are available on request from the corresponding author.

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
