# Peer review of "Mortality Risk and Urinary Proteome Changes in Acute COVID-19 Survivors in the Multinational CRIT-COV-U Study"

_biomedicines, 2024, doi:10.3390/biomedicines12092090_

Round 1

Reviewer 1 Report

Comments and Suggestions for Authors

The manuscript entitled "Mortality risk and urinary proteome changes of acute COVID-19 survivors in the multinational study CRIT-COV-U study" is good piece of work which is important in the current post covid scenario.

1. Words such as "we" can be removed from the manuscript as its not a part of scientific literature

2. In methods section of abstract, authros can mention specific peptides

3. Conclusion under abstract section remains vague; specific findings and recommendations are needed under this section.

4. In the introduction, authors need to mention about the most important secondary and long lasting effects of COVID, which may have resulted in analyzing the mortality aspects 

5. I could find hardly 3-4 references in Introduction section; I recommend to enrich the literature support.

6. The results and discussion secions are well drafted with clear communication and supported by graphics, tables and supplementary materials.

7. There are several typographic issues present (Eg: Line 46: Spelling of "consequence"). Authors need to correct such issues

Comments on the Quality of English Language

1. There are several typographic issues present (Eg: Line 46: Spelling of "consequence"). Authors need to correct such issues

Author Response

Reviewer: 1

Comments to the Authors

  1. Words such as "we" can be removed from the manuscript as its not a part of scientific literature

Response: We thank the reviewer for this very helpful comment. We agree with reviewer suggestion and have revised the manuscript accordingly, ensuring it aligns with the conventions of scientific literature.

  1. In methods section of abstract, authors can mention specific peptides

Response: We thank the reviewer for this advice. To address reviewer suggestion, we included now the following information on specific protein fragments in the abstract:

“These peptides are fragments of albumin, alpha-2-HS-glycoprotein, apolipoprotein A-I, beta-2-microglobulin, CD99 antigen, various collagens, fibrinogen alpha, polymeric immunoglobulin receptor, sodium/potassium-transporting ATPase, and uromodulin and were integrated these into a predictive classifier (DP201).”

  1. Conclusion under abstract section remains vague; specific findings and recommendations are needed under this section.

Response: We also thank the reviewer for this comment. Both comments 2 and 3 have helped very much to  improve the abstract. We have revised the conclusion of the abstract as follows:

The peptide-based classifier demonstrated significant predictive value for mortality in post-acute COVID-19 patients, highlighting the utility of urinary peptides in prognosticating post-acute COVID-19 mortality, offering insights for targeted interventions. By utilizing these defined biomarkers in the clinic, risk stratification, monitoring, and personalized interventions may be significantly improved. Our data also suggest that mortality should be considered as one possible symptom or consequence of post-acute sequelae of SARS-CoV-2 infection, a fact that is currently overlooked.

  1. In the introduction, authors need to mention about the most important secondary and long lasting effects of COVID, which may have resulted in analyzing the mortality aspects 

Response: We thank the reviewer for his/her valuable feedback. We have revised the introduction to include a discussion of the most important secondary and long-lasting effects of COVID-19. These effects, often referred to as Long COVID, have been linked to ongoing physiological stress and organ damage, which may contribute to an increased risk of mortality among survivors. We have integrated this context to better explain our focus on analyzing mortality aspects in the study. We hope this revision addresses your comment effectively.

  1. I could find hardly 3-4 references in Introduction section; I recommend to enrich the literature support.

Response: We thank the reviewer for also for this valuable advice. We have enriched the Introduction section by adding additional references. We hope these additions strengthen the literature support and address the comment effectively.

  1. The results and discussion secions are well drafted with clear communication and supported by graphics, tables and supplementary materials.

Response: We thank the reviewer for this positive comment.

  1. There are several typographic issues present (Eg: Line 46: Spelling of "consequence"). Authors need to correct such issues

Response: We thank the reviewer for pointing out the typographic errors. We apologize for the oversight and have carefully revised the manuscript to correct all typographic issues, including the spelling mistake on line 46.

Reviewer 2 Report

Comments and Suggestions for Authors

This is a study on urine biomarkers associated with prognosis in COVID-19 infection and is a multicentre study involving six European countries. The study shows that certain prognostic predictions can be made by analysing peptides in patients' urine. It is a unique and large scale trial and is potentially worth for publication.

This study is related to Reference 5 study and overlaps in terms of study team, cohort and methodology. Therefore, it is important to introduce the content of reference 5 and explain the significance of this study based on its results. However, this reviewer thinks  there is lack of sufficient explanation and it is difficult to understand the position of this paper in relation with reference 5.

The study shows that mortality is high after the acute phase - a period known as Long COVID. However, it is difficult to know to what extent deaths in the chronic phase (e.g. after 3 months of illness) are increasing, as the analysis seems to include deaths in the acute and sub-acute phases (The Kaplan-Meyer curve in Figure 1A seems to present the data, but it is a light-coloured figure, making it difficult to discern).

The conclusion of this paper, the discovery of biomarkers associated with death in chronic phase, seems to be insufficiently supported by the data.

The Crit-Cov-U-study-protocol states on page 8 that 'A urine sample will be obtained from each patient with COVID-19 on Day 0-1, Day 4-5, and Day 10-14. '

However, it is not clear when the samples were collected for the current study.

Author Response

Reviewer2:

Comments to the Author

  1. This study is related to Reference 5 study and overlaps in terms of study team, cohort and methodology. Therefore, it is important to introduce the content of reference 5 and explain the significance of this study based on its results. However, this reviewer thinks there is lack of sufficient explanation and it is difficult to understand the position of this paper in relation with reference 5.

Response: We thank the reviewer for this comment. To address this, we have expanded the Introduction to provide further details on the content of Reference 5, highlighting its key findings within the acute phase of COVID-19. We have also clarified that our study investigated mortality after the acute phase, and therefore builds upon and differs from the previous research. We hope this additional information helps to clearly position our paper in relation to Reference 5 and enhances the overall understanding of our study's contribution.

  1. The study shows that mortality is high after the acute phase - a period known as Long COVID. However, it is difficult to know to what extent deaths in the chronic phase (e.g. after 3 months of illness) are increasing, as the analysis seems to include deaths in the acute and sub-acute phases (The Kaplan-Meyer curve in Figure 1A seems to present the data, but it is a light-coloured figure, making it difficult to discern).

Response: We appreciate the reviewer’s valuable feedback. In our study, we focused exclusively on patients who survived the acute phase of COVID-19, specifically analyzing data with a follow-up period of more than three weeks. The acute phase (and not the sub-acute phase) was excluded from our analysis. To address the reviewer's concerns, we have made several revisions within the text. Additionally, we have included a zoomed-in version of Figure 1a to demonstrate that no deaths that occurred during the acute phase were taken into account and to provide additional clarity regarding the phases analyzed. This should make it easier to discern the Kaplan-Meier curve data and confirm that our analysis pertains to the post-acute period, where we observe the increased mortality associated with Long COVID. We hope these revisions improve the clarity and comprehensibility of our findings.

  1. The conclusion of this paper, the discovery of biomarkers associated with death in chronic phase, seems to be insufficiently supported by the data.

Response: We appreciate the reviewers’ observation and have taken steps to clarify our focus on the post-acute phase of COVID-19. As previously mentioned, our analysis specifically targets the post-acute phase, not only chronic, and we have revised the manuscript to consistently use this term throughout. We hope these revisions address your concerns and provide a clearer and more robust foundation for our findings.

The Crit-Cov-U-study-protocol states on page 8 that 'A urine sample will be obtained from each patient with COVID-19 on Day 0-1, Day 4-5, and Day 10-14. '

However, it is not clear when the samples were collected for the current study.

Response: We thank the reviewer for bringing this to our attention. In our study, we utilized only the first urine sample collected from participants in the Crit-Cov-U study. We have now included this specific detail in the Methods section to ensure clarity regarding the timing of the sample collection: “Urine peptide data of first urine samples collected within 3 days of a positive PCR were used”

Round 2

Reviewer 1 Report

Comments and Suggestions for Authors

No more comments.

Author Response

Reviewer: 1

Comments to the Author

No comments

Response: Thank you very much for your positive opinion.

Reviewer 2 Report

Comments and Suggestions for Authors

The manuscript of the paper has been significantly improved.

The lines of the Kaplan-Meier curve are difficult to distinguish. If possible, colouring or other measures should be taken to improve this. 

Author Response

Reviewer2:

Comments to the Author

The manuscript of the paper has been significantly improved.

The lines of the Kaplan-Meier curve are difficult to distinguish. If possible, colouring or other measures should be taken to improve this. 

Response: We thank the reviewer for the value suggestion. We have addressed the concern by revising Figure 1 and applying different colors to the Kaplan-Meier curves, making them easier to distinguish. In addition to using color, we have ensured that the line styles are distinct to enhance clarity, particularly for those viewing the figure in black and white or for those with color vision deficiencies. We believe these improvements will make the data presentation clearer and more accessible.
